# Selective Oxofunctionalization of Cyclohexene over Titanium Dioxide-Based and Bismuth Oxyhalide Photocatalysts by Visible Light Irradiation

**Adolfo Henríquez [1]** [ID]**, Héctor D. Mansilla [2], Azael Martínez-de la Cruz [3], Lorena Cornejo-Ponce [1], Eduardo Schott [4,5] and David Contreras [2,5,*]** [ID]

[1]  Departamento de Ingeniería Mecánica, Facultad de Ingeniería, Universidad de Tarapacá,
     Avda. General Velásquez 1775, Arica 1000007, Chile; adohenriquez@udec.cl (A.H.);
     lorenacp@academicos.uta.cl (L.C.-P.)
[2]  Facultad de Ciencias Químicas, Universidad de Concepción, Concepción 4070386, Chile; hmansill@udec.cl
[3]  Faculty of Mechanic and Electric Engineering, Autonomous University of Nuevo León, University Village,
     San Nicolás de los Garza 66451, Nuevo León, Mexico; azael70@gmail.com
[4]  Departamento de Química Inorgánica, Facultad de Química y Farmacia, Centro de Energía UC,
     Centro de Investigación en Nanotecnología y Materiales Avanzados CIEN-UC,
     Pontificia Universidad Católica de Chile, Avenida Vicuña Mackenna 4860, Santiago 7820436, Chile;
     edschott@uc.cl
[5]  ANID-Millennium Science Initiative Program-Millennium Nuclei on Catalytic Process towards Sustainable
     Chemistry (CSC), Avda. Vicuña Mackenna 4860, Macul, Santiago 8970117, Chile
\*   Correspondence: dcontrer@udec.cl; Tel.: +56-41-2203431; Fax: +56-41-2207310

**Abstract:** Photocatalysis driven under visible light allows us to carry out hydrocarbon oxofunctionalization under ambient conditions, using molecular oxygen as a sacrificial reagent, with the absence of hazardous subproducts and the potential use of the Sun as a clean and low-cost source of light. In this work, eight materials—five based on titanium dioxide and three based on bismuth oxyhalides—were used as photocatalysts in the selective oxofunctionalization of cyclohexene. The cyclohexane oxofunctionalization reactions were performed inside of a homemade photoreactor equipped with a 400 W metal halide lamp and injected air as a source of molecular oxygen. In all assayed systems, the main oxygenated products, identified by mass spectrometry, were 1,2-epoxycyclohexane, 2-cyclohexen-1-ol, and 2-cyclohexen-1-one. Titanium dioxide-based materials exhibited higher selectivities for 1,2-epoxycyclohexane than bismuth oxyhalide-based materials. In addition to this, titanium dioxide doped with iron exhibited the best selectivity for 1,2-epoxycyclohexane, demonstrating that iron plays a relevant role in the cyclohexene epoxidation process.

**Keywords:** titanium dioxide; bismuth oxyhalide; cyclohexene; photocatalysis; oxofunctionalization

## 1. Introduction

The heterogeneous photocatalysis of semiconductors is an advanced oxidation process (AOP) based on the ability of photocatalysts to adsorb photons with an energy equal to or greater than the forbidden energy band of the semiconductor material ($h\nu \geq E_{bg}$). The electron transfer from the valence band of the photocatalyst generates a photohole-photoelectron pair [1] with oxidizing and reducing properties, respectively. The generated redox potentials are high, and there are often hydroxyl radicals, hydrogen peroxide and other reactive oxygen species (ROS), that are generated from the redox transformation of water. Due to the high reactivity of these systems, they have mainly been applied in the elimination of pollutants and disinfection of water [1–7]. Due to their low selectivity, these systems have been rarely applied in the selective conversion of organic compounds [8].



Many semiconductor materials have been used as catalysts in photocatalytic processes. Nevertheless, it is generally accepted that titanium dioxide-based materials are the most appropriate for conducting photocatalytic processes due to their high photoreactivity, chemical and biological stability, photostability, and low cost [9]. It has been frequently reported that transition metal ions enhance the photocatalytic activity of titanium dioxide by suppressing charge carrier recombination and also by onset shifting in the absorption band gap to the visible region [10]. Among the transition metals used as doping elements, iron(III) has attracted special attention because iron cations greatly influence the charge-carrier recombination time. The presence of iron(III) induces a bathochromic effect, and using an iron-doped photocatalyst has proven to be efficient in several important photocatalyzed reduction and oxidation reactions [11]. Furthermore, the ionic radius of iron(III) (0.79 Å) is similar to that of titanium(IV) (0.75 Å), so iron(III) can easily be substituted for titanium(IV) in the $TiO_2$ lattice [12,13].

The main disadvantage of titanium dioxide-related photocatalysts is their wide bandgap, close to 3.2 eV, which limits the use of these materials in photocatalytic processes under an electromagnetic radiation of under 400 nm. To allow for the efficient use of sunlight radiation as an energy source in photochemical processes, many efforts have been focused on the development of new semiconductor materials such as metal sulfides, metal oxides, and modified oxides such as bismuth oxyhalides (BiOX, X = Cl, Br, or I). BiOX materials have been recently used as photocatalysts because they exhibit photocatalytic activity under visible light, as well as adequate mechanical and chemical stability and optical and electrical properties [14]. On the other hand, to enhance the photoactivity of titanium dioxide under irradiation with higher wavelengths, the forbidden energy band of titanium dioxide has been adjusted [8,15,16].

The selective aerobic oxidation of hydrocarbons is a commercially important process to obtain oxygenated products, such as alcohols, ketones, and carboxylic acids, and epoxides [17]. Nevertheless, the processes performed to obtain these kinds of products are frequently severe, require corrosive mixtures of chemical reagents, and are non-selective [18]. Photocatalysis potentially allows for the oxidation of hydrocarbons under room conditions [19] using visible radiation sources (e.g., solar radiation) as a clean and low-cost energy source [20]. In this context, the rational use of appropriate semiconductor materials with photocatalytic activity and the fine control of chemical reactions may lead to highly selective organic transformations by photocatalytic oxofunctionalization [21].

Recently, different AOP systems have been utilized for the photocatalytic oxofunctionalization of hydrocarbons [22]. In these systems, the carbon-oxygen bonds are generated from the activation of carbon-hydrogen bonds catalyzed by a semiconductor material with photocatalytic activity. Among the reported oxofunctionalization reactions, we may mention the conversion of alkanes to alcohols, aldehydes, ketones, and carboxylic acids; the oxidation of alcohols to the respective aldehydes and ketones; and the hydroxylation of aromatic compounds [23]. Special attention has been placed on the selective synthesis of epoxides (epoxidation) because of their high value for synthesis in the fine chemical industry and as building blocks [24,25].

Green strategies for epoxide obtention (in room conditions) have been developed, mainly through homogeneous catalysis by iron complexes [26,27]. Nevertheless, these organic ligands are expensive, toxic, air/moisture sensitive, and commercially unavailable [28,29].

There have been a few papers that have included selective cyclohexene epoxidation by heterogeneous photocatalysts, and most of these are based on $TiO_2$ (Table 1). These systems show selectivities ranging from 1.2% to 26% under UV radiation due to the band gap of $TiO_2$. To our knowledge, there have been no studies under visible (or solar-like radiation) using photocatalysts with a lower energy band gap like BiOX.

**Table 1.** Reported studies on the selective photocatalytic oxidation of cyclohexene over TiO$_2$-based photocatalysts.

| Reference | Catalyst | Illumination Source | Experimental Conditions | Time (h) | Conversion (%) | Epoxide Selectivity (%) |
|---|---|---|---|---|---|---|
| [30] | TiO$_2$ | ($\lambda > 280$ nm) | Catalyst, 10 mg; MeCN, 10 mL; cyclohexene, 0.2 mmol; O$_2$, 1 atm. | 3 | 9 | 26 |
| [31] | Degussa P25 TiO$_2$ | 125 W Hg lamp (>340 nm) | Catalyst, 15 mg; cyclohexene, 20 mL; O$_2$, 100 mL min$^{-1}$; room temperature. | 1 | 5 | 18 |
| [32] | Degussa P25 TiO$_2$ | 125 W Hg lamp ($\lambda > 300$ nm) | Cyclohexene, 10 mL; room temperature | 3 | - | 9 |
| [33] | TiO$_2$ | ($\lambda > 365$ nm) | Catalyst, 4 mg mL$^{-1}$ in the O$_2$-saturated neat cyclohexene. | 3–4 | - | 1.17 |

In the present work, photocatalytic systems driven by visible radiation were assayed for the selective synthesis of 2-epoxycyclohexane from cyclohexene using TiO$_2$- and BiOX-based photocatalysts in order to obtain a better selectivity in this new green method for 2-epoxycyclohexene synthesis.

## 2. Results

### 2.1. Photocatalyst Characterization

The X-ray diffraction data of the titanium dioxide-based photocatalysts (Table 2) were in agreement with Joint Committee on Powder Diffraction Standards (JCPDS) Card No. 4-477, corresponding to the anatase phase of titanium dioxide. On the other hand, the diffraction peaks of the bismuth oxychloride (BiOCl), bismuth oxybromide (BiOBr), and bismuth oxyiodide (BiOI) photocatalyst samples (Table 3) could be indexed to the tetragonal phases of BiOCl (JCPDS Card No. 6-249), BiOBr (JCPDS Card No. 78-348), and BiOI (JCPDS Card No. 73-2062), respectively. Figure 1 shows the DR-UV-vis (diffuse-reflectance ultraviolet-visible) spectra of the synthesized photocatalysts. There exists an optimum value for the iron amount due to the recombination charge increasing with the dopant concentration; this is because the distance between trapping sites in a particle decreases with the number of dopants. On the other hand, if the iron content is very low, there are fewer trapping sites available, thus reducing the activity of the photocatalyst [34]. In Figure 1A, it can be observed that titanium dioxide doped with iron had the ability to absorb photons of higher wavelengths in comparison to undoped titanium dioxide. This shift increased depending on the Fe proportion.

In Figure 1B, it can be observed that nitrogen used as a dopant enhanced the ability of undoped titanium dioxide to absorb photons in the visible region. On the other hand, the spectra of the bismuth oxyhalides present a dependence on the ionic radius of the halogen for the shifting of the bandgap to a higher wavelength. Nitrogen doping enhanced the photocatalytic activity of the titanium dioxide-based materials due to mixing of the nitrogen p states with oxygen 2p states, contributing to bandgap narrowing. In 2001, Asahi et al. reported that nitrogen-doped titanium dioxide exhibits a higher photocatalytic activity under visible light in the photodegradation of methylene blue and gaseous acetaldehyde (wavelength of <500 nm) [35]. Controversially, it has been proposed that the nitrogen precursor during the modification procedure induces color centers associated with oxygen vacancies, which are themselves responsible for visible light activity [36,37].

Figure 1C shows the spectra of the bismuth oxyhalide-based photocatalysts. Bismuth oxyhalides crystalized in layered structures comprising [Bi$_2$O$_2$]$^{2+}$ slabs interleaved with double halide atom layers. These materials have attracted great attention in research due to their electric, magnetic, and optical properties [38]. Based on density functional theory calculations, it was determined that the valence band of bismuth oxyhalides was dominated by oxygen 2p states and halogen np states (where *n* is 3, 4,

and 5 for chlorine, bromine, and iodine, respectively), and the conduction band was dominated by bismuth 6p states. With an increasing atomic number, the density peak of halogen np shifts towards the top of the valence band and narrows the bandgap [39]. The absorption maximum of the synthesized bismuth oxyhalides (Figure 1C) shifted towards higher wavelengths. This shift increased with a dependence on the increasing atomic number, which was in concordance with the literature [39,40]. More information about the synthesis and characterization of the photocatalysts was presented in the work of Henríquez et al. in 2017 [41].

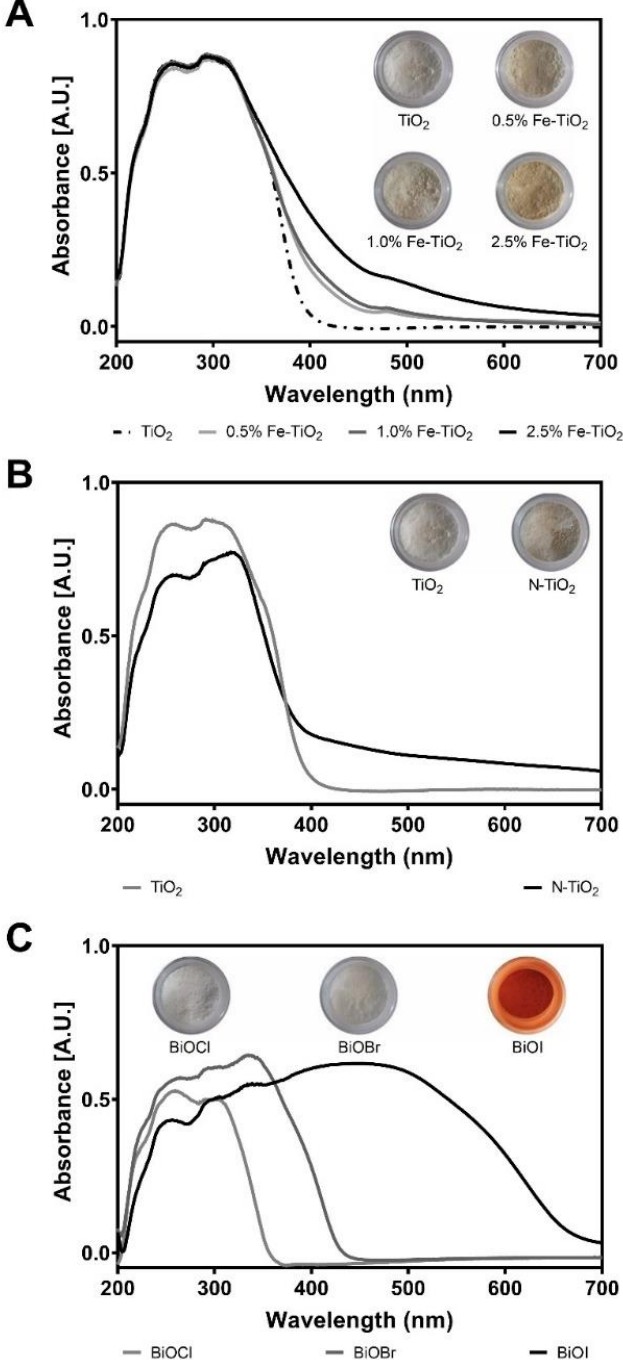

**Figure 1.** Diffuse reflectance spectra of titanium dioxide and bismuth oxyhalide photocatalysts: (**A**) iron-doped titanium dioxide, (**B**) nitrogen-doped titanium dioxide, and (**C**) bismuth oxyhalide photocatalysts. Insets show pictures of photocatalyst samples.

**Table 2.** XRD data of the photocatalysts based on titanium dioxide.

| (h k l) Planes | JCPDS Card No. 4-477 2θ (Deg) | I | TiO$_2$ 2θ (Deg) | I | 0.5% Fe TiO$_2$ 2θ (Deg) | I | 1.0% Fe TiO$_2$ 2θ (Deg) | I | 2.5% Fe TiO$_2$ 2θ (Deg) | I | N TiO$_2$ 2θ (Deg) | I |
|---|---|---|---|---|---|---|---|---|---|---|---|---|
| 1 0 1 | 25.16 | 100 | 25.40 | 100 | 25.39 | 100 | 25.35 | 100 | 25.34 | 100 | 25.34 | 100 |
| 0 0 4 | 37.63 | 22 | 37.96 | 32 | 37.95 | 31 | 37.91 | 30 | 37.90 | 32 | 37.98 | 30 |
| 2 0 0 | 47.96 | 33 | 48.13 | 36 | 48.12 | 34 | 48.11 | 34 | 48.06 | 34 | 48.06 | 36 |
| 1 0 5 | 53.71 | 21 | 54.00 | 28 | 53.98 | 26 | 53.95 | 25 | 53.94 | 26 | 54.01 | 24 |
| 2 1 1 | 55.07 | 19 | 55.26 | 23 | 55.26 | 22 | 55.24 | 21 | 55.21 | 17 | 55.00 | 23 |
| 2 0 4 | 62.57 | 13 | 62.79 | 21 | 62.76 | 20 | 62.72 | 19 | 62.69 | 18 | 62.73 | 16 |
| 1 1 6 | 68.52 | 5 | 68.91 | 10 | 68.92 | 9 | 68.87 | 9 | 68.81 | 8 | 68.97 | 4 |
| 2 2 0 | 70,27 | 5 | 70.44 | 9 | 70.44 | 8 | 70.42 | 7 | 70.37 | 7 | 70.36 | 5 |
| 2 1 5 | 74.95 | 10 | 75.21 | 12 | 75.14 | 11 | 75.19 | 10 | 75.15 | 10 | 75.26 | 9 |

JCPDS: Joint Committee on Powder Diffraction Standards.

**Table 3.** XRD data of the photocatalysts based on bismuth oxyhalides.

| (h k l) Planes | JCPDS Card No. 6-249 2θ (Deg) | I | BiOCl 2θ (Deg) | I | (h k l) Planes | JCPDS Card No. 78-348 2θ (Deg) | I | BiOBr 2θ (deg) | I | (h k l) Planes | JCPDS Card No.73-2062 2θ (Deg) | I | BiOI 2θ (Deg) | I |
|---|---|---|---|---|---|---|---|---|---|---|---|---|---|---|
| 0 0 1 | 11.89 | 40 | 11.88 | 6 | 0 0 1 | 10.80 | 43 | 10.83 | 7 | 0 0 2 | 19.28 | 4 | 19.382 | 5 |
| 0 0 2 | 23.94 | 16 | 24.01 | 3 | 0 0 2 | 21.75 | 7 | 21.89 | 3 | 0 1 1 | 24.21 | 2 | 24.34 | 5 |
| 1 0 1 | 25.77 | 100 | 25.92 | 68 | 1 0 1 | 25.12 | 25 | 25.24 | 34 | 0 1 2 | 29.66 | 100 | 29.69 | 100 |
| 1 1 0 | 32.45 | 75 | 32.64 | 100 | 1 0 2 | 31.63 | 100 | 31.74 | 73 | 1 1 0 | 31.64 | 42 | 31.74 | 54 |
| 1 0 2 | 33.40 | 95 | 33.48 | 38 | 1 1 0 | 32.20 | 48 | 32.34 | 100 | 1 1 1 | 33.18 | 2 | 33.22 | 5 |
| 1 1 1 | 34.72 | 10 | 34.85 | 6 | 1 1 1 | 34.02 | 3 | 34.18 | 5 | 0 1 3 | 37.09 | 9 | 37.05 | 6 |
| 1 1 2 | 40.78 | 30 | 41.01 | 11 | 1 1 2 | 39.27 | 11 | 39.39 | 9 | 1 1 2 | 37.41 | 5 | 37.46 | 6 |
| 2 0 0 | 46.58 | 35 | 46.81 | 38 | 2 0 0 | 46.18 | 19 | 46.36 | 34 | 0 0 4 | 39.33 | 7 | 39.37 | 6 |
| 2 0 1 | 48.28 | 10 | 48.46 | 5 | 1 1 3 | 46.80 | 8 | 46.97 | 5 | 0 2 0 | 45.42 | 17 | 45.49 | 24 |
| 1 1 3 | 49.61 | 25 | 49.69 | 9 | 2 0 1 | 47.55 | 3 | 47.74 | 4 | 1 1 4 | 51.44 | 14 | 51.40 | 12 |
| 2 0 2 | 53.08 | 8 | 53.32 | 3 | 1 0 4 | 50.57 | 12 | 50.68 | 5 | 1 2 2 | 55.22 | 28 | 55.228 | 30 |
| 2 1 1 | 53.96 | 25 | 54.28 | 23 | 2 1 1 | 53.37 | 6 | 53.49 | 8 | 1 1 5 | 60.28 | 4 | 60.105 | 3 |
| 2 1 2 | 58.50 | 30 | 58.73 | 16 | 1 1 4 | 56.05 | 11 | 56.14 | 5 | 0 2 4 | 61.75 | 6 | 61.644 | 4 |
| 1 1 4 | 60.46 | 12 | 60.57 | 4 | 2 1 2 | 57.13 | 29 | 57.25 | 26 | 1 2 4 | 66.43 | 5 | 66.273 | 6 |
| 2 2 0 | 67.96 | 12 | 68.34 | 7 | 2 2 0 | 67.46 | 5 | 67.62 | 6 | 0 3 2 | 74.31 | 5 | 74.196 | 4 |
| 3 0 1 | 74.02 | 6 | 74.44 | 3 | 2 1 4 | 71.00 | 7 | 70.97 | 3 | | | | | |
| 2 1 4 | 74.84 | 14 | 75.03 | 2 | 3 1 0 | 76.70 | 6 | 76.89 | 5 | | | | | |

## 2.2. Cyclohexene Oxofunctionalization

The cyclohexene oxofunctionalization reactions were performed in a homemade photoreactor equipped with a 400 W metal halide lamp (Osram, Powerstar HQI-E 400W/D Pro Daylight, Munich, Germany). The spectral distribution of the metal halide lamp, reported in the product datasheet, is shown in Figure 2, and the band gap energies of the photocatalysts assayed are represented by vertical lines over the spectrum. It should be noted that BiOI had the lowest bandgap and was more efficient in using the radiation emitted by the lamp. On the contrary, BiOCl was the compound with the highest bandgap; therefore, it was the least efficient compound in taking advantage of the radiation from the lamp. As is discussed later, this difference in spectral behavior was not directly related to selectivity in the oxidation of cyclohexene (Table 4).

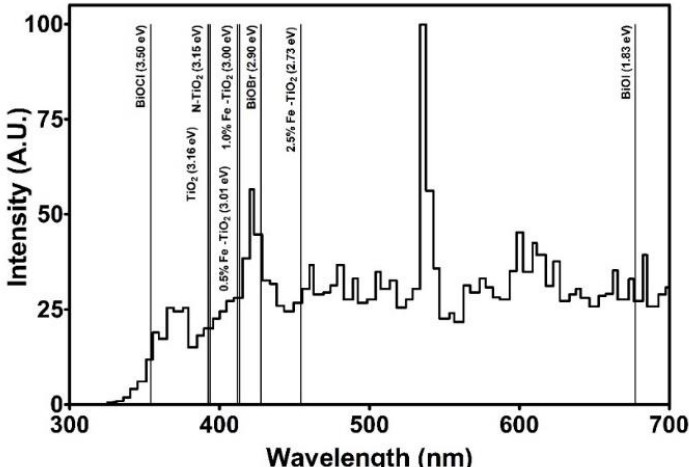

**Figure 2.** The emission spectrum of the Osram Powerstar HQI-E 400 W/D Pro Daylight lamp (adapted from the spectral distribution reported in the lamp's datasheet). Vertical lines represent the bandgaps of titanium dioxide- and bismuth oxyhalide-based photocatalysts.

**Table 4.** Product selectivity of the photocatalytic oxofunctionalization of cyclohexene over the synthesized photocatalysts.

| Entry | Photocatalyst | Time [h] | Temperature [°C] | Selectivity (%) | | | |
|---|---|---|---|---|---|---|---|
| | | | | A | B | C | D |
| 1 | None | 3 | 37 | 5.7 | 2.8 | 85.1 | 6.4 |
| 2 | $TiO_2$ | 3 | 37 | 10.0 | 14.8 | 69.4 | 5.8 |
| 3 | 0.5% Fe-$TiO_2$ | 3 | 37 | 16.9 | 26.9 | 47.2 | 9.0 |
| 4 | 1.0% Fe-$TiO_2$ | 3 | 37 | 20.4 | 35.4 | 36.3 | 7.9 |
| 5 | 2.5% Fe-$TiO_2$ | 3 | 37 | 13.3 | 22.2 | 59.6 | 4.9 |
| 6 | N-$TiO_2$ | 3 | 37 | 11.0 | 16.3 | 62.6 | 10.1 |
| 7 | BiOCl | 3 | 37 | 6.2 | 7.0 | 79.0 | 7.8 |
| 8 | BiOBr | 3 | 37 | 6.8 | 8.0 | 80.8 | 4.4 |
| 9 | BiOI | 3 | 37 | 3.3 | 0.2 | 88.9 | 7.6 |

Reaction conditions: 1 bar of air, 24.975 mL (0.247 mol) of cyclohexene, 0.025 mL (1.39 mmol) of water, 25 mg of the photocatalyst, and magnetic stirring. Selectivity according to Equation (3) (%): A: 1,2-epoxycyclohexane; B: 2-cyclohexen-1-ol; C: 2-cyclohexen-1-one; and D: other byproducts.

The cyclohexene oxofunctionalization was performed in the reaction system shown in Figure 3. In this system, 24.975 mL of cyclohexene was placed in a 50 mL two-necked round-bottomed flask equipped with a reflux condenser in the presence of 25 mg of the respective catalyst (1 mg/mL) and 25 μL of nanopure water (55.5 mmol $L^{-1}$). In each case, during the reaction time, the reaction system was magnetically stirred and saturated by air bubbling (1 atm) as a source of molecular

oxygen. Considering the solubility of $O_2$ in cyclohexane [42], its concentration was about 2.6 mmol $L^{-1}$. To avoid an increase in pressure inside the reaction system, the reflux condensers were capped with a rubber stopper pierced through by a hollow needle. The temperatures inside the photocatalytic reactor reached 37 ± 2 °C.

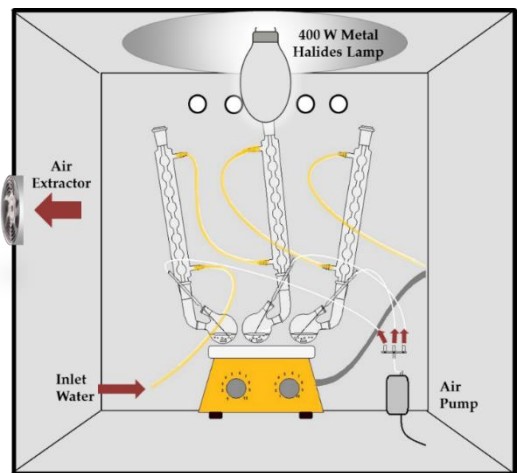

**Figure 3.** Scheme of the photocatalytic reactor.

### 2.3. Identification of the Oxofunctionalized Products

After a reaction time of 180 min, the products 1,2-epoxycyclohexane, 2-cyclohexen-1-ol, and 2-cyclohexen-1-one were identified in all photocatalytic systems assayed by gas chromatography-mass spectrometry (GC-MS). The matching of the experimental mass spectrum obtained for each product of cyclohexene oxofunctionalization with the reference mass spectrum in the NIST/EPA/NIH (National Institute of Standards and Technology/Environmental Protection Agency/National Institutes of Health) Mass Spectral Library (NIST 05) is shown in Figure 4. The areas under the chromatographic peaks obtained by gas chromatography-flame ionization detection (GC-FID) were integrated and are represented in Figure 5. All the chromatographic peaks of non-identified compounds were integrated, and the areas under these peaks were summed and presented like the others. In both the GC-MS and GC-FID systems, the analytes were separated in a crosslinked 5% diphenyl, 95% dimethylsiloxane column, and the temperature program was an isotherm set at 40 °C.

The selectivities were estimated as the ratio between the peak area of the product and the sum of the areas of peaks of all products generated in the oxofunctionalization reaction, according to Equation (1):

$$Selectivity\ (\%) = \frac{A_{\text{product}}}{A_{1,2-\text{epoxycyclohexane}} + A_{2-\text{cyclohexen}-1-\text{ol}} + A_{2-\text{cyclohexen}-1-\text{one}} + A_{other}} \qquad (1)$$

where $A_{\text{product}}$, $A_{1,2\text{-epoxycyclohexane}}$, $A_{2\text{-cyclohexen-1-ol}}$, $A_{2\text{-cyclohexen-1-one}}$, and $A_{others}$ correspond to the integrated peak areas of oxofunctionalized products 1,2-epoxycyclohexane, 2-cyclohexen-1-ol, 2-cyclohexen-1-one, and other by-products, respectively.

The cyclohexene suffered oxidation in the absence of photocatalyst at the assayed conditions. This was in agreement with a report by Mahajani et al. in 1999 [43]. The selectivities of the oxofunctionalization of cyclohexene under visible light by the prepared photocatalysts are shown in Figure 3. The main product in all assayed systems was 2-cyclohexen-1-one. The selectivity of the conversion of cyclohexene to 1,2-epoxycyclohexane was higher over titanium dioxide-based photocatalysts than over bismuth oxyhalide photocatalysts (Figure 5 and Table 4). Among the titanium dioxide-based photocatalysts, higher selectivities for 1,2-epoxycyclohexane were achieved by

iron-doped titanium dioxide photocatalysts. These results indicated that iron, when used as a dopant, plays an important role in the epoxidation of cyclohexene,

Based on the results discussed above, the proposed pathways for the oxidation of cyclohexane using the TiO$_2$-based photocatalysts and bismuth oxyhalide photocatalysts are shown in Figure 6. For all studied photocatalysts, in the first stage, H• abstraction by the photohole gave C$_6$H$_9$•. Then, this radical reacted with molecular oxygen to give C$_6$H$_9$OO•. This compound disproportionated to 2-cyclohexen-1-ol (C$_6$H$_9$OH) and 2-cyclohexen-1-one (C$_6$H$_8$O). On the other hand, the absorbed C$_6$H$_9$OH was further oxidized to C$_6$H$_8$O, resulting in lower levels of C$_6$H$_9$OH than C$_6$H$_8$O. A minor quantity of adsorbed C$_6$H$_9$OO• reacted with adsorbed C$_6$H$_{10}$ molecules to give 1,2-epoxycyclohexane (C$_6$H$_8$O). Over iron-doped titanium dioxide photocatalysts, the epoxidation of adsorbed cyclohexene molecules occurred, thus decreasing the formation of C$_6$H$_9$OO• and, subsequently, C$_6$H$_9$OH and C$_6$H$_8$O.

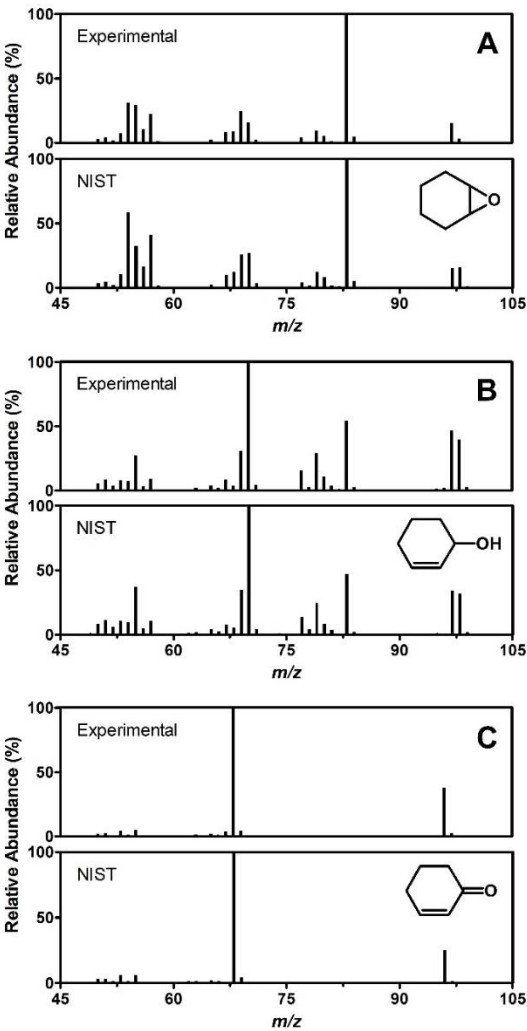

**Figure 4.** Comparison of the mass spectra of products obtained from oxofunctionalization of cyclohexene under visible light irradiation with the mass spectra of epoxycyclohexane, 2-cyclohexen-1-ol, and 2-cyclohexen-1-one reported in the National Institute of Standards and Technology (NIST) 05 database: (**A**) 1,2-epoxycyclohexane, (**B**) 2-cyclohexen-1-ol, and (**C**) 2-cyclohexen-1-one.

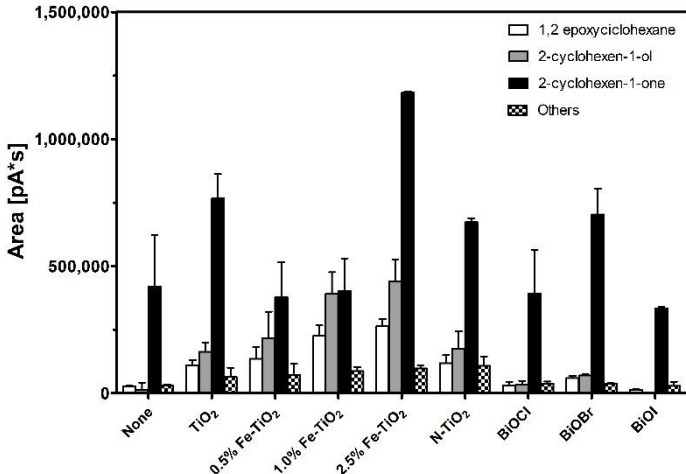

**Figure 5.** The yield of oxofunctionalized cyclohexene under visible light irradiation. White, gray, black, and checkered columns represent yields of epoxycyclohexane, 2-cyclohexen-1-ol, 2-cyclohexen-1-one, and others, respectively.

**Figure 6.** Proposed pathways for the oxofunctionalization of cyclohexene over titanium dioxide- and bismuth oxyhalide-based photocatalysts.

## 3. Discussion

The development of photocatalytic activity under visible light and the fine control of reaction conditions may promote selective organic transformations at room temperature and atmospheric pressure, the use of molecular oxygen as an oxidizing agent, and the potential use of sunlight as a clean and low-cost energy source [20]. In the present work, a reaction system was constructed with cyclohexene as the solvent. The $H_2O$ concentration was 55.5 mmol/L, and that of $O_2$ was about 2.6 mmol/L (considering the constant air bubbling and approximating the solubility of $O_2$ in cyclohexane) [42], with 25 mg of the respective photocatalyst. Denekamp et al. in 2018 [44] reported the catalytic selective oxidation of cyclohexene with molecular oxygen. In their study, the best selectivity for 1,2-epoxycyclohexane was obtained for copper oxide supported on N-doped carbon, where the selectivity for 1,2-epoxycyclohexane was 15%. Their reaction conditions were 70 °C, 10 bar of $O_2$, 2.5 mL of cyclohexene, 0.5 mL of cyclohexane, 10 mg of the catalyst, 15 mL of acetonitrile, 1 mL of hydrogen peroxide, and magnetic stirring. However, in the absence of hydrogen peroxide,

the selectivity for epoxide decreased to 9%. This observation demonstrated the relevance of radical species generated by hydrogen peroxide decomposition in the cyclohexene epoxidation mechanism. In the present study, the best photocatalytic epoxidation performance of cyclohexene was observed for 1.0% Fe-TiO$_2$, where the selectivity for epoxide was 20.4% (Entry 4, Table 4)

In Figure 5, it can be observed that the obtained yields of oxofunctionalized products in the systems catalyzed by bismuth oxyhalides were non-significantly different to those obtained in an uncatalyzed system. This behavior was due to the low specific area of bismuth oxyhalide-based photocatalysts in comparison to titanium dioxide-based photocatalysts. On the other hand, the titanium dioxide-based photocatalysts exhibited better performance in the oxofunctionalization of cyclohexane under visible light irradiation. This may be explained by the ability of TiO$_2$ photocatalysts to reduce O$_2$ to H$_2$O$_2$ [45] under light activation.

Non-significant differences were observed in the yields of oxofunctionalized products in the system catalyzed by nitrogen-doped titanium dioxide in comparison to the system catalyzed by undoped titanium dioxide. A dependence of oxofunctionalized product yields was observed in systems photocatalyzed by iron-doped titanium dioxide photocatalysts. The yields of oxofunctionalized products increased with increasing atomic percentage of iron in the titanium dioxide. Thus, in systems that include iron-doped titanium dioxide photocatalysts, in-situ-generated H$_2$O$_2$ can interact with Fe to generate a peroxo complex. This is supported by the results described by McAteer et al. in 2013 [46], who reported the catalytic oxidation of cyclohexene catalyzed by iron(III)/hydrogen peroxide (H$_2$O$_2$) in mildly acidic solution. The authors found that in the absence of molecular oxygen, the reaction yields of cyclohexen-1-ol and cyclohexen-1-one were significantly reduced in comparison to that of epoxide. The authors suggested that an activated form of H$_2$O$_2$ over Fe is responsible for epoxide formation, rather than organoperoxy radicals, which have been reported to generate epoxides from alkenes but in very small yields.

Considering the antecedents mentioned above, we propose that the higher yields of epoxide in systems photocatalyzed by iron-doped titanium dioxide are due to the formation of an iron(III)-peroxo complex in the surface of these photocatalysts (Figure 7).

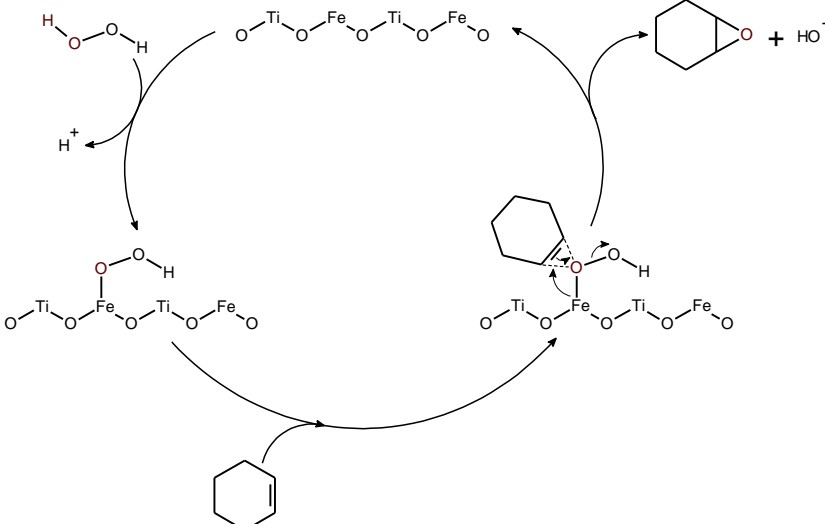

**Figure 7.** Proposed pathway for the epoxidation of cyclohexene over iron-doped titanium dioxide photocatalysts.

## 4. Materials and Methods

### 4.1. Synthesis of Photocatalysts

In the present work, previously synthesized and characterized photocatalysts were used. A complete description of the synthesis method and the characterization are presented in the work of Henriquez et al. in 2017 [41]. Iron-doped titanium dioxide photocatalysts were synthesized according to a modified synthetic procedure described by Qamar et al. in 2014 [47]. Nitrogen-doped titanium dioxide was synthesized according to a solvothermal-assisted sol-gel synthetic procedure described by Zalas in 2014 [48]. Bismuth oxyhalides (BiOCl, BiOBr, and BiOI) were prepared according to a modified synthetic procedure described by Wang et al. in 2015 [39].

### 4.2. Photocatalyst Characterization

The X-ray diffraction patterns of the synthesized photocatalysts were used to investigate the phase structure of the materials. On the other hand, the crystallite sizes of the photocatalysts were estimated using the Debye-Scherrer formula (Equation (2)), where $D$ is the average crystallite size, $K$ is a dimensionless shape factor (in this case, with a value of 0.94), $\lambda$ is the X-ray radiation wavelength (Cu K$\alpha$ = 0.154056 nm), $\beta$ is the band broadening at half the maximum intensity (FWHM), and $\theta$ is the diffraction angle.

$$D = \frac{K\,\lambda}{\beta\cos\theta} \tag{2}$$

The bandgap of the semiconductor photocatalyst was determined from a Tauc plot obtained from the UV-vis diffuse reflectance spectra. The relational expression proposed by Tauc et al. in 1966 [49] and Davis and Mott in 1970 [50] (Equation (3)) was used.

$$h\nu\alpha^{1/n} = A\left(h\nu - E_g\right) \tag{3}$$

where $h$ is Planck's constant, $\nu$ is the vibration frequency, $\alpha$ is the absorption coefficient, $E_g$ is the bandgap, and $A$ is a proportional constant. The value of the exponent $n$ denotes the nature of the simple transition. For an indirectly allowed transition, $n = 2$. The $\alpha$ term in the Tauc equation was substituted with the Kubelka-Munk function, F(R$_\infty$). The crystal morphology of the products was observed by scanning electron microscopy (NanoSEM 200, FEI-Nova, Hillboro, OR, USA). Nitrogen adsorption isotherms at −196 °C were obtained using a BELSORP-mini II surface area and pore size analyzer.

### 4.3. Cyclohexene Oxofunctionalization

The oxofunctionalization of 25 mL of cyclohexene was performed in a 50 mL 2-neck round-bottom flask fitted with a reflux condenser in the presence of 25 mg of the catalyst (1 mg/mL) and 25 μL of nanopure water. For each reaction, the system was saturated with 1 atm of air for the duration of the reaction time (3 h). Photocatalytic oxidation reactions were carried out under visible radiation generated by a 400 W metal halide lamp (Osram Powerstar HQI-E 400 W/D Pro Daylight, Munich, Germany). The temperature inside the photocatalytic reactor reached 37 ± 2 °C.

To identify the compounds obtained from the photocatalytic oxidation of cyclohexane, a 0.2 μL aliquot was removed from the reaction mixture and injected into a GC-MS system. GC-MS analysis was performed on a 5890 Series II gas chromatograph (Hewlett Packard Corporation, Palo Alto, CA, USA) interfaced with a 5972 Mass Selective Detector Quadrupole (Hewlett Packard Corporation, Palo Alto, CA, USA). The analytes were separated in a crosslinked 5% diphenyl, 95% dimethylsiloxane column (Hewlett Packard Corporation, Palo Alto, CA, USA). The temperature program was an isotherm at 40 °C for 15 min. Helium was used as the carrier gas at a constant flow of 1 mL min$^{-1}$, and data acquisition was performed in electron impact ionization (EI) mode. The temperature inlet was set to 250 °C, and the source was set to 280 °C. In another analysis, an aliquot of 0.2 μL taken from the reaction mixture was injected into a Series II gas chromatograph coupled to a flame ionization detector

(Hewlett Packard Corporation, Palo Alto, CA, USA). The analytes were separated in a crosslinked 5% diphenyl, 95% dimethylsiloxane column (Hewlett Packard Corporation, Palo Alto, CA, USA). The temperature program was an isotherm set at 40 °C for 10 min. Nitrogen was used as the gas carrier at a constant flow of 1 mL/min. The temperature inlet was set to 250 °C, and the detector was set to 180 °C. The data acquisition and peak integration of the oxofunctionalized products from cyclohexene were performed using the HP 3398A GC Chemstation Software (version A.01.01, Hewlett-Packard, Palo Alto, CA, USA, 1998).

## 5. Conclusions

Based on obtained results, it is possible to conclude that titanium dioxide-based materials exhibit better photocatalytic performance than bismuth oxyhalide-based materials on the photocatalytic oxofunctionalization of cyclohexene under visible light irradiation. On the other hand, the highest selectivity for 1,2-epoxycyclohexane exhibited by titanium dioxide doped with iron was attributed to the ability of these materials to generate hydrogen peroxide in situ and the subsequent formation of an iron(III)-peroxo complex in the surface of these photocatalysts that promotes the epoxidation process.

**Author Contributions:** Conceptualization, D.C. and H.D.M.; methodology, A.M.-d.l.C. and A.H.; software, A.M.-d.l.C. and A.H.; validation A.H.; formal analysis, A.H.; investigation, A.H.; resources, D.C.; writing—original draft preparation, A.H. and L.C.-P.; writing—review and editing, L.C.-P., A.H., E.S., and D.C.; visualization, A.H.; supervision, H.D.M.; project administration, D.C.; funding acquisition, D.C. All authors have read and agreed to the published version of the manuscript.

**Funding:** FONDECYT 1201895; Millennium Science Initiative of the Ministry of Economy, ANID—Millennium Science Initiative Program—NCN17_040, and FONDAP Solar Energy Research Center SERC-Chile 15110019.

**Conflicts of Interest:** The authors declare no conflict of interest. The funders had no role in the design of the study; in the collection, analyses, or interpretation of data; in the writing of the manuscript; or in the decision to publish the results.

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
