# Peer review of "Selective Oxofunctionalization of Cyclohexene over Titanium Dioxide-Based and Bismuth Oxyhalide Photocatalysts by Visible Light Irradiation"

_catalysts, doi:10.3390/catal10121448_

Round 1

Reviewer 1 Report

The manuscript reported a work on "different photocatalyst based on titanium dioxide and bismuth oxyhalides". Frankly, extensive investigations have been conducted. Both abstract and introduction were well-written, however, there are some unclear points as listed below. Thus, this will be recommended after adequate revision. The comments are following:

1) English style should be improved, there are many unclear/mistakes point such as:

Line 40: “photohole-photoelectron pair[1]”—it should be “photohole-photoelectron pair [1]”; same status: Line 42, Line 44, Line 49, Line 56, Line 61, Line 63, Line 65, Line 69, Line96,…

2) Writing proficiency should be improved, For English writing, the punctuation mark is important for helping reader to understand the content of report. Same unclear/mistakes were appeared at Line 109, Line 123, Line 178 and Line 227, please improve them.

3) For reference mark, many different styles were appeared in this manuscript, such as: “ketones and carboxylic acids and epoxides[16]”, “above the valence band edge (Evb). [28]” and “reducing the activity of photocatalyst.[32]”, please improve them.

4) About “et al”, at some part, the italic type was used, and some part were not. Please improve them.

5) About “figure”, sometimes “figure” without capitalizing the first letter, sometimes with capitalizing the first letter, such as in line 127, line 130, line 140, line 148, line 172 and line 180. Please improve them.

6) For the Figure 1A, it’s difficult to differentiate the 1.0% Fe-TiO2 and 2.5% Fe-TIO2 from the figure, please change another line style or color.

7) For the Figure 2, what’s the mean of “A”, ”B” and ”C” in figures?

8) For Table 1, what’s the mean of “A”, ”B”, ”C” and “D”?

9) As author mentioned, the XRD (X-ray diffraction patterns) was used for characterized the sample, however, please show the XRD data for samples.

10) Photocatalytic activity tests of samples were carried out under the visible radiation, please show the visible range.

11) Furthermore, I would like to know if the sample was characterized by the other measurements or test? Such as XPS (X-ray photoelectron spectroscopy) and specific surface area (SSA, BET-method or other method), for helping reader to understand the different between samples.

Author Response

Q1. The manuscript reported a work on "different photocatalyst based on titanium dioxide and bismuth oxyhalides". Frankly, extensive investigations have been conducted. Both abstract and introduction were well-written, however, there are some unclear points as listed below. Thus, this will be recommended after adequate revision. The comments are following:

English style should be improved, there are many unclear/mistakes point such as:

R1. Thank you for review our paper, you are right, we did a lot of mistakes, apologize for it. We repair these and sent the paper for a professional English Editing Services at the website of the MDPI Author Services.

Q2. Line 40: “photohole-photoelectron pair[1]”—it should be “photohole-photoelectron pair [1]”; same status: Line 42, Line 44, Line 49, Line 56, Line 61, Line 63, Line 65, Line 69, Line96,…

R2. We repair these mistakes.

Q3. Writing proficiency should be improved, For English writing, the punctuation mark is important for helping reader to understand the content of report. Same unclear/mistakes were appeared at Line 109, Line 123, Line 178 and Line 227, please improve them.

R3. We are sorry for this evident mistake, we repair this.

Q4. For reference mark, many different styles were appeared in this manuscript, such as: “ketones and carboxylic acids and epoxides[16]”, “above the valence band edge (Evb). [28]” and “reducing the activity of photocatalyst.[32]”, please improve them.

R4. We check all the reference marks and homologize the format, thank you for realizing about that.

Q5. About “et al”, at some part, the italic type was used, and some part were not. Please improve them.

R5. We check all the “et al” and homologize the format, in italics. Thank you for realizing about that.

Q6. About “figure”, sometimes “figure” without capitalizing the first letter, sometimes with capitalizing the first letter, such as in line 127, line 130, line 140, line 148, line 172 and line 180. Please improve them.

R6. We check all the “figure” and homologize the format, with the first capital letter. Thank you for realize about that.

Q7. For the Figure 1A, it’s difficult to differentiate the 1.0% Fe-TiO2 and 2.5% Fe-TIO2 from the figure, please change another line style or color.

R7. We change the black and withe lines by colors in order to clarify this.

Q8. For the Figure 2, what’s the mean of “A”, ”B” and ”C” in figures?

R8. We include letters A, B and C in the captions for clarify this.

Q9. For Table 1, what’s the mean of “A”, ”B”, ”C” and “D”?

R9. We apologize, because the table caption was missed. We include this in this new version.

Q10. As author mentioned, the XRD (X-ray diffraction patterns) was used for characterized the sample, however, please show the XRD data for samples.

R10. The XRD profiles of photocatalysts samples were previously report in Henríquez et al 2017. A few days ago, we asked for permission to use figures to Elsevier and we are awaiting their response. Nevertheless, in the new version of manuscript we include a Table that compare the XRD peaks of our photocatalysts samples with JDPDS cards, in order to demonstrate the crystalline structure of synthesized materials.

Q11. Photocatalytic activity tests of samples were carried out under the visible radiation, please show the visible range.

R11. Thank you for your comments. In this version of manuscript, we include the reported spectrum of metal halide lamp used as light source in the photocatalytic experiments. The spectrum of this lamp measured in our laboratory was shown as supplementary material in a previous work (doi:10.1016/j.apcatb.2017.01.022.) cited in this paper (reference 39).

Q12. Furthermore, I would like to know if the sample was characterized by the other measurements or test? Such as XPS (X-ray photoelectron spectroscopy) and specific surface area (SSA, BET-method or other method), for helping reader to understand the different between samples.

R12. The samples of photocatalysts were characterized by SEM, BET-method, DR-UV-Vis and XRD in order to determine their morphology, specific area, bandgap and crystalline phase, respectively. A Table that summarize the characterization of samples is include in the new version of manuscript. The characterization of these materials is showed in more detail in a previous report (doi:10.1016/j.apcatb.2017.01.022.) cited in this paper (reference 27).

Reviewer 2 Report

This paper describes the oxidative function of cyclohexane by BiOX compounds and visible light.  I think there is a publishable paper in here but the current manuscript is not ready.

I could not write a paper in any language but my own, so I am always impressed by anything at all readable written by non-native speakers.  That said, there are significant issues of clarity that need to be worked out here.  These are not just small errors that don't affect the authors' ability to get across their intent.  I literally do not understand what they mean in some places.  In others, it just needs a good cleaning up.

The authors should also rethink what they include in what section of this paper.  The introduction, for example spends time talking about apparently completely unrelated material (roughly lines 80-90).  Material in the first part of the Results section appears to be more introduction, in that it doesn't seem to be reporting anything actually done by the authors.

As the authors point out, these BiOX compounds have been around for some time now.  Although I don't follow this as closely as I once did, is this really the first study talking about their use in this general field of semi-selective oxidations?  That should be addressed more thoroughly in the introduction.  I couldn't tell if they were telling me their BiOX compounds were the same as everyone else's (as per the UV spectra) or different...and if so, how.

The Results section has a major flaw in that I have no idea how the photolysis were carried out. (It is in the experimental section, but should be summarized in results for context.)  The light source would appear to generate UV...is that filtered out??? Why is there water? What's the light source?  The single detail of the length of irradiation tells us literally nothing.  The term "A" in equation 3 tells us nothing about whether the instrument response factors are equivalent...which they should be if this is really a selectivity.

I have no idea how much conversion is achieved.  The utility of this as a method is modest if we are talking about converting only a few percent of the cyclohexene to oxygenated products, particularly since, as noted, no catalyst is required to get some reasonably selective oxidation to cyclohexenone.

I don't really know what I am to take away from this.  Arguably, the paper might conclude that there's really not much need for any of these catalysts unless an argument can be made for wanting to produce a mixture containing cyclohexenol.

The proposed mechanism is more of a pathway and cursory at best.  There is zero chance that a cyclohexenyl radical will abstract an OH hydrogen.  That reaction is rather endothermic.

The discussion (!) section tells me a little more about the experimental conditions that should be in the results.

From this report, I don't mean to sound so negative that I don't think this can be saved.  I think this IS a reasonably interesting topic.  But the paper isn't well organized, and I am not clear what is novel nor what the take-away really should be.  Once those things are clarified, it's likely that this could be a reasonable paper.

Author Response

Q1. This paper describes the oxidative function of cyclohexane by BiOX compounds and visible light.  I think there is a publishable paper in here, but the current manuscript is not ready.

I could not write a paper in any language but my own, so I am always impressed by anything at all readable written by non-native speakers.  That said, there are significant issues of clarity that need to be worked out here.  These are not just small errors that don't affect the authors' ability to get across their intent.  I literally do not understand what they mean in some places.  In others, it just needs a good cleaning up.

R1. We realize about the problems in this version of the manuscript, we hope that now, could be ready for publication. We realize, about the problems in the English quality and the organization of the manuscript. We reorganized the paper, include more information about the material characterization, and figure that include the lamp spectra and the band gap of the catalysts. In addition, we include a scheme of the reactor. On the other hand, the English of this new version was edited by the MIDI AG English language editing service. We hope that now, our manuscript could be easier to be understood.

Q2. The authors should also rethink what they include in what section of this paper.  The introduction, for example spends time talking about apparently completely unrelated material (roughly lines 80-90).  Material in the first part of the Results section appears to be more introduction, in that it doesn't seem to be reporting anything actually done by the authors.

R2. Thank you for your comments. We rewrite the manuscript according to your suggestions and observations. We eliminate the paragraph where we mentioned other systems used for epoxidation, but that are not related to photocatalysts. In addition, we include more information in the results section and move some general sentences to the introduction section.

Q3. As the authors point out, these BiOX compounds have been around for some time now.  Although I don't follow this as closely as I once did, is this really the first study talking about their use in this general field of semi-selective oxidations?  That should be addressed more thoroughly in the introduction.  

R3. We include a more complete revision of literature about that and we add a table with literature data for  cyclohexene selective oxidation about TiO2 related compounds. We did not found literature about cyclohexene selective oxidation by BiOX related photocatalysts

Q4.  I couldn't tell if they were telling me their BiOX compounds were the same as everyone else's (as per the UV spectra) or different...and if so, how.

R4. We utilized BiOX compounds previously synthetized and characterized. We clarify this in more detail in addition to the reference mention.

Q5 The Results section has a major flaw in that I have no idea how the photolysis were carried out. (It is in the experimental section, but should be summarized in results for context.)  

R5. We include more detailed explanation in the results section in order to introduce the results. In addition we include an scheme of the reaction system in a new scheme (Figure 3). In the new version of manuscript, we include a paragraph in both, Results and Experimental sections with information about physical and optic properties of synthesized materials used as photocatalysts.

Q6. The light source would appear to generate UV...is that filtered out???

R6. We include a figure with the emission spectra of the lamp to clarify this issue. (figure 2).

Q7. Why is there water?

R7. The water is necessary for carried out the reaction, because, this is adsorb in the surface of the photocatalyst and the OH radicals are generated. We include this sentence in the results section.

Q8.  What's the light source?  The single detail of the length of irradiation tells us literally nothing.  

R8. We include a figure with the emission spectra of the lamp to clarify this issue. (figure 2). In addition we include the technical characteristic of the lamp.

Q9. The term "A" in equation 3 tells us nothing about whether the instrument response factors are equivalent...which they should be if this is really a selectivity.

R9. The term A correspond to area under curve of chromatographic peaks of each compound. In this work samples were analyzed by GC equipped with a flame ionization detector (FID). The response factor of FID detector is proportional to the number of carbon atoms in a hydrocarbon molecule. Due to all the identified compound generated in the photocatalytic process contain 6 carbons atoms, a similar response of this detector in expected for these molecules,

Q10. I have no idea how much conversion is achieved.  The utility of this as a method is modest if we are talking about converting only a few percent of the cyclohexene to oxygenated products, particularly since, as noted, no catalyst is required to get some reasonably selective oxidation to cyclohexenone.

I don't really know what I am to take away from this.  Arguably, the paper might conclude that there's really not much need for any of these catalysts unless an argument can be made for wanting to produce a mixture containing cyclohexenol.

R10. In absence of catalysts the selectivity for 1,2-epoxyxlyclohexane and 2-cyclohexen-1-ol are 5.7 % and 2.8 %, respectively. In presence of titanium dioxide doped with iron at 1.0 at % these selectivities arise to 20.4 % and 35.4 %. These observations demonstrate that the selectivities are modulated by effect of photocatalyst. Similar effect is observed with other photocatalysts based on titanium dioxide used in the present work. On the other hand, a little effect on the selectivity on oxofunctionalization of cyclohexene is observed with photocatalysts based on bismuth oxyhalides. Due to high reactivity of cyclohexene, it is easily oxidized by molecular oxygen in absence of photocatalysts and the main effect of photocatalyst are observed on selectivity instead on yields of reaction.

Q11. The proposed mechanism is more of a pathway and cursory at best.  There is zero chance that a cyclohexenyl radical will abstract an OH hydrogen.  That reaction is rather endothermic.

R11. We agree with you, the figure was modified, the mentioned step was deleted from proposed mechanism of cyclohexene photooxidation.

Q12. The discussion (!) section tells me a little more about the experimental conditions that should be in the results.

R12. Thank you for your comments. We rewrite the section 4.3 (cyclohexene oxofunctionalization) of the manuscript according to your suggestions and observations. In this version of the manuscript we include a paragraph where we mentioned the experimental conditions in the section of results.

Q13. From this report, I don't mean to sound so negative that I don't think this can be saved.  I think this IS a reasonably interesting topic.  But the paper isn't well organized, and I am not clear what is novel nor what the take-away really should be.  Once those things are clarified, it's likely that this could be a reasonable paper.

R13. We greatly appreciate your comments and suggestions. All of them have contributed to present an improved version of the manuscript.